# Formation of Point Defects Due to Aging under Natural Conditions of Doped GaAs

**DOI:** 10.3390/ma17061399

**Published:** 2024-03-19

**Authors:** Samuel Zambrano-Rojas, Gerardo Fonthal, Gene Elizabeth Escorcia-Salas, José Sierra-Ortega

**Affiliations:** 1Grupo de Investigación en Física del Estado Sólido, Universidad de la Guajira, Rihoacha 440007, Colombia; 2Grupo de Investigación en Teoría de la Materia Condensada, Universidad del Magdalena, Santa Marta 470004, Colombia; geneescorcias@unicesar.edu.co; 3Instituto Interdisciplinario de las Ciencias, Universidad del Quindio, Armenia 630001, Colombia; gfonthal@uniquindio.edu.co; 4Grupo de Óptica e Informática, Departamento de Física, Universidad Popular del Cesar, Sede Hurtado, Valledupar 200001, Colombia

**Keywords:** point defects, vacancies, aging dynamics, natural conditions, doped GaAs

## Abstract

The aging dynamics of materials used to build the active part of optoelectronic devices is a topic of current interest. We studied epitaxial samples of GaAs doped with Ge and Sn up to $1\times 10^{19}$ cm^−3^, which were stored in a dry and dark environment for 26 years. Photoluminescence spectra were taken in three periods: 1995, 2001 and 2021. In the last year, time-resolved photoluminescence, Raman, and X-ray measurements were also performed to study the evolution of defects formed by the action of O_2_ in the samples and its correlation with the doping with Ge and Sn impurities. We found that oxygen formed oxides that gave off Ga and As atoms, leaving vacancies mainly of As. These vacancies formed complexes with the dopant impurities. The concentration of vacancies over the 26 years could be as large as $1\times 10^{18}$ cm^−3^.

## 1. Introduction

In recent decades, some theoretical and experimental studies have shown a growing interest in the analysis of the effects of aging and the degradation rate in semiconductor materials [1,2], and luminescent ceramics [3]. The purpose of these studies is to understand, simulate, and mitigate their impact during the design phase of devices for technological and industrial applications, as described by Kim and his group in a review of the year 2023 [4]. Defects are often responsible for device degradation. Even in the absence of degradation, defects can limit device performance [5]. The aging dynamic can affect the speed, power consumption, and functionality of digital circuits, such as microprocessors, memory devices, and logic gates. Compensation by point defects can decrease the doping level that can be achieved. Defects with energy levels within the band gap can act as recombination centers, which prevents carrier collection, for example, in a solar cell [6] or in a high electron mobility transistor (HEMT) [7]. Luminescence centers can be used as light emitters at specified wavelengths; or single spin centers that can act as an artificial atom and serve as a qubit in a quantum information system [8]. For this reason, materials with many defects are deliberately grown. Examples of these are materials for ultrafast optoelectronic switches with pulses of the order of THz, where the defect density must be such that the carrier lifetime is as short as a few picoseconds [9]. In many laboratories, semiconductor wafers are either purchased or manufactured and then stored on shelves in ambient conditions for several years. However, it is often not realized that there is a slow internal dynamic within the solid material that can alter its physical properties over time.

Nowadays, GaAs-based microelectronic devices are widely used in consumer and defense technology. Several devices have been built with GaAs and its alloys: heterojunction bipolar transistor (HBT), quantum cascade laser (QCL), high-speed electronic devices, and even biosensors [10]. Consequently, at present, there is a deep knowledge of its electronic structure and physical, optical and electrical parameters. For this reason, we have selected this material to study the aging of semiconductors, since its physical parameters are well known and we can make comparisons. Several techniques have been tested to accelerate the aging process and be able to study it in the laboratory.Irradiation with protons [11], exposure to an environment of H_2_O and O_2_ for 36 months [12], application of current for hours [13], and even the recent creation of a new aging test [14] provide insight into the significant emphasis currently placed on understanding the aging of optoelectronic materials.

For all of the above reasons, we consider it convenient to conduct a comprehensive study of the aging dynamics in semiconductor materials and their relationship with impurities at different doping levels. In this work, we studied the aging dynamics over three time periods when GaAs epitaxial films were exposed to the environment for 26 years and their relationship with impurities in the range from light to heavy doping with Ge and Sn. We conducted the study using powerful techniques such as photoluminescence (PL), time-resolved photoluminescence (TRPL), and other specialized techniques such as X-ray diffraction (XRD) and Raman spectroscopy. We present interesting results that show the formation of vacancies in both As and Ga, at concentrations as high as $1\times 10^{18}$ cm^−3^, and impurity-vacancy complexes that cause greater deviations from the lattice parameter.

## 2. Materials and Methods

In 1995 GaAs:Sn and GaAs:Ge thick films were grown on a GaAs substrate oriented in the (100) direction by Liquid Phase Epitaxy and at a temperature of 800 °C. The materials and their purity were Ga(7N), Ge(5N), Sn(5N) and GaAs to obtain As [15]. As can be seen, the resulting films will contain native V*_As_* and interstitial Ga defects due to an excess of Ga [16]. The samples were thermally annealed at 620 °C for 4 h in a H_2_ environment to facilitate crystalline ordering and avoid deep defects or traps. The films were built up to an average thickness of 10 μm. The Ge concentrations ranged from $1\times 10^{16}$ cm^−3^ to $1\times 10^{19}$ cm^−3^ and the Sn concentrations ranged from $1\times 10^{17}$ cm^−3^ to $2\times 10^{18}$ cm^−3^, which were measured by the capacitance-voltage method. After photoluminescence and photoreflectance measurements, the samples were stored for six years in a dry and dark environment. In 2001, Fonthal [17] also measured photoluminescence and photoreflectance and proposed that the variations in these spectra were due to new vacancies added to the films by the reactivity of O_2_ to form Ga_2_O_3_ and As_2_O_3_, the latter being volatile. He proved the hypothesis by chemically pickling part of the samples and obtaining the original spectra measured in 1995. Figure 1 shows this result.

The samples were stored again in a dark and dry environment for another 20 years. In 2021, photoluminescence (PL), Raman, X-ray (RX) and time-resolved photoluminescence (TRPL) spectra were taken. The photoluminescence system used to perform the spectroscopic measurements of the semiconductor emission of GaAs described in this work uses a laser with a wavelength of 488 nm focused on the sample by a cylindrical lens as the external excitation source. A convergent lens in the entrance slit of a Horiba FHR1000 spectrophotometer (Horiba Co., Kyoto, Japan) focused the light emission from the sample under study. To remove the Rayleigh radiation reflected by the sample, a LP’488 filter was located in the entrance slit. The spectrophotometer has a diffraction grating of 1800 lines per millimeter, a focal distance of 100 cm, a spectral resolution of 0.010 nm and as photodetector, a CCD camera with thermo-electric cooling.

The measures of relaxation times of the photoluminescence signal were performed in a PicoQuant time-resolved photoluminescence system, using as the excitation source, a pulsed laser diode PicoQuant LDH-P-C-450 (PICOQUANT, Berlin, Germany). This laser has tunable excitation power and frequency, and the laser control system is synchronized to the TimeHarp 260 (PICOQUANT, Berlin, Germany) board used for the signal processing. The FWHM of the laser emission is less than 100 ps. The measurements were acquired using 80 MHz and less than 1.2 mW of the laser radiation. The detection system used is the PMA 182 and has a photomultiplier tube with thermo-electric cooling, a high voltage source, and a preamplifier. The time response of the PMA 182 is 180 ps (FWHM). The voltage signal is sent to the TimeHarp 260 board. The processing of the signal is performed in a TimeHarp 260 PICO Single TCSPC (Time–Correlated Single Photon Counting System). Considering the sparseness of the collected photons, the data collection is performed over multiple cycles of excitation and emission. Each point of the photoluminescence decay profile is a histogram of the single photon events collected over many cycles. The PicoQuant company software allows us to adjust the zero time of the start of the laser pulse with the start of the count of emitted photons.

For the application of the Raman spectroscopy technique, high-resolution Horiba model H R Evolution micro Raman equipment was used. A 488.0 nm Ar+ laser was used as the excitation source; the equipment has a thermoelectrically cooled CCD detector. To obtain the measurements, the laser signal was focused with a 100× objective, which allowed a lateral resolution of approximately 1 μm to be obtained. X-ray diffraction patterns were obtained on a diffractometer (Rigaku, ultima IV) with a detector D/tex ultra, operating at 35 kV and 15 mA, with a Cu Kα radiation wavelength of 1.5406 Å, and from 5 to 70° on a 2Θ scale with a step size of 0.02°. All measurements by the different techniques were performed at room temperature (25 ± 1 °C).

## 3. Results and Discussions

In Figure 2, the photoluminescence results for the undoped GaAs sample used by Torres [15] in 1995 are depicted in black. In 2001, Fonthal [17] utilized the same sample, which was stored in dark conditions, outdoors, in an average temperature of 25 °C. Fonthal’s spectrum is represented in red. This same sample was employed for this study and its spectrum is illustrated in blue. It is evident that there has been a change in the optical response of the undoped GaAs sample over time due to aging, at least within the laser penetration length of the PL technique. In these spectra, two zones can be distinguished: a zone above 1.375 eV where excitonic peaks appear near to the energy gap at 1.424 eV and a second peak around 1.385 eV that corresponds to the donor-acceptor transition of shallow impurities, in this case Sn and Ge. It can be distinguished that in the blue spectral curve, the excitonic peak broadens and shifts towards higher energies, while the intensity of the donor-acceptor transition decreases compared to the spectra of lower aging ages. This is due to the deterioration of the material’s crystalline quality over time. The zone below 1.375 eV shows a broad peak that corresponds to transitions involving deep defects, such as vacancies. In this second zone, it can be seen that the defect peak shifts towards lower energy values, widens and increases in intensity with respect to the excitonic peak, as the sample ages. This is due to the increase in defect concentration in samples as years go by. The change in energy of the defect peak will be explained later in this section.

With the TRPL technique, variations in the photoresponse over time were constructed for a GaAs sample at 300 K, with one of the cases being shown in the Figure 3. The sample was excited with a 405 nm laser pulsed at 80 MHz. The laser pulse terminates in 1.125 ns, and the internal processes within the semiconductor samples subsequently de-excite, generating the signal depicted in the graph.

The settings for time decay were determined using a biexponential function, with the R^2^ value serving as the adjustment criterion. In all cases, the R^2^ was greater than 0.998. When a monoexponential function was tested, the value of R^2^ was very low, and with a triexponential, one of the time values of the biexponential function was repeated. Table 1 displays the relaxation times and their associated errors for different samples.

From Table 1, it can be seen that the results for t_1_ are not statistically different, and the same applies for t_2_. Therefore, it can be stated that the type of trap must be the same, regardless of the degree of impurity concentration. However, the relaxation times for undoped GaAs are slightly different from those doped with impurities, which means that because the impurity occupied the Ga or As sites in GaAs, the vacancies of these elements are not formed, and therefore, the electron takes longer to fall into the traps, because there are fewer deep traps. Table 1 also shows that as the concentration of impurities increases, the percentage participation of the smallest time transition increases while that of the longest time decreases.

The 405 nm laser energy generates electron-hole pairs, which de-excite by the electron returning from the conduction band to the valence band within a time of a few picoseconds after a femtosecond thermalization process within the conduction band [18]. In a time of tens of ps, excitons radiate as bonded electron-hole pairs that coexist for a certain period of time [19]. These two time frames cannot be observed with our TRPL system. Between 200 and 1500 picoseconds, de-excitations to traps are measured, primarily when dealing with donor–acceptor pairs (DA), that manifest as impurity-vacancy complexes in terms of defects. The percentage participation of Table 1 supports this idea; that is, for undoped GaAs, there are many vacancies and few complexes; but as the concentration of impurities increases, complexes are formed, and that is why the participation of t_1_ is greater for higher concentrations.

In our case, the impurities are Ge and Sn, and the vacancies can be either Ga or As. Vashistha et al. found times ranging from 1.25 ns to 1.38 ns for local complex defects [20]. Oh et al. found times of 900 ps for the DV*_Ga_* complex, with D being a positive donor [21]. In 2017, Elsayed and Krause-Rehberg found a time of 290 ps for Zn*_Ga_*−V*_As_* [22]. Based on this, the value of t_1_ obtained by us must correspond to the transition between I*_Ga_*−V*_As_*, where the letter I represents impurities. We interpret the longer time t_2_ as a transition between the valence band and the V*_As_*. These times should be longer than the previous ones because a phonon must participate in this transition due to the tetragonal distortion caused by the vacancy. It is less likely for three “particles” (electron, photon, and phonon) to come together than just the photon and electron. In Figure 2, it can be seen in the defect zone that the peak has structure, which corresponds to phononic replicas, demonstrating phononic participation in this type of transition. Niemeyer et al. measured times ranging from 5.2 ns to 7.0 ns in GaAs:Zn for different concentrations of Zn, associating them with transitions to defect states not specified in the paper [23].

Figure 4a shows the X-ray diffractograms for the undoped GaAs sample and four doped samples. The first information that can be gleaned from these diffractograms is that all five samples are indeed GaAs, as confirmed by comparison with PDF 800016. Figure 4b reveals that, for each colored peak representing the diffractogram of a different sample, there is a shift towards larger angles as the doping concentration increases. In addition, each peak of the (400) line has a smaller satellite on the side of larger angles, which corresponds to the response for the K$\alpha_{2}$ line, λ = 1.5444 Å. These lines are equally separated from the main line for each sample.

Through the Bragg diffraction law, the lattice parameters *a* were calculated using the angle Θ, the K$\alpha_{1}$ laser line of λ = 1.5406 Å and the separation between planes with the lattice parameter and the Miller indices for a cubic lattice. Table 2 shows the lattice parameters calculated in this way. In addition, the change in the perpendicular parameter due to tetragonal distortion was calculated using the equation from references [24,25]. The grain size D was also calculated using the Scherrer equation.

The lattice constant *a* for the undoped sample is slightly larger than that reported by Qadri et al. at 5.6515 Å [26] due to the fact that the vacancies widen the lattice. For doped samples, the a constant decreases as the impurity concentration increases. This is because the impurity compensates for the sites of native vacancies in GaAs, which widen the lattice due to the Janh-Teller distortion effect. The change in atomic radii due to impurities is not very significant, since Ge replaces As (r*_Ge_* = 1.22 Å, r*_As_* = 1.18 Å) and Sn replaces Ga (r*_Sn_* = 1.40 Å, r*_Ga_* = 1.26 Å). Therefore, the decrease in the lattice parameter when increasing the concentration of impurities is better explained by the change in vacancies. The grain sizes are roughly equal, indicating that differences in a diffractogram are not due to grain boundaries. Figure 5 shows our values of $△a_{⊥}/a$ as a function of the impurity concentrations. These values are compared with those obtained by De Lyon et al. [24] and Bhunia et al. [27], including their theoretical curves, both for GaAs:C. We note that our values are higher than theirs, despite the fact that C, like Ge and Sn, is also amphoteric. The difference is due to the fact that the impurity concentrations measured by Torres do not include the concentration of defects that have emerged in recent years. Two data points from Lanyi et al. [28] are also included for GaAs:Si, which is also amphoteric, and show the change in the sample’s $△a_{⊥}/a$ value before and after annealing. The deformation is higher in the oxidized sample than in the clean one, as occurs with our samples. Figure 5 suggests that the actual concentration should be the sum of impurities and defects, which could exceed $10^{18}$ cm^−3^.

In Figure 2, it can be seen in the spectrum obtained by Torres that there is a band of defects centered at 1.368 eV, just after the sample was grown. This band corresponds to what is known as native defects, which, in the case of GaAs, have been reported by several authors [29,30] as V*_As_*− I*_As_*. Fonthal [17] managed to demonstrate that V*_Ga_*− I*_Ga_* are also formed. The formation of Ga and As oxides due to the presence of oxygen adds defects to the surface slowly and over many years, as suggested by Birey and Site [31], Lum and Wiedel [32], and Bunea and Dunham [33]. The same graph shows the PL spectrum obtained by Fonthal six years later, where it can be seen that the defect zone has expanded and shifted towards lower energies, centered at 1.335 eV. This means that new defects have formed in greater quantity. For this work, we measured the PL spectrum 26 years after the undoped GaAs sample was grown and found that the defect zone had grown significantly more and was located at 1.240 eV. Xu and Lindefelt [34] found the different charge states in both V*_Ga_* and V*_As_*, as shown in Table 3.

We believe that the variation in energy of the defect zone over the years, as shown in Figure 2, is due to a change in the charge state of vacancies rather than the emergence of new defects. This change in the charge state of the vacancy is attributed to the self-compensation process in semiconductors [35,36]. From Table 3, it can be inferred that as both the negative and positive charge states decrease, the energy of the trap, in this case, the vacancy, increases. A simple subtraction between the energy gap (1.424 eV) and the energy position of the maximum intensity of the defect zone of Torres, Fonthal, and ourselves, gives energy differences of 0.056 eV, 0.089 eV, and 0.184 eV, respectively. This rough calculation yields values very similar to those reported by Xu and Lindefelt, shown in Table 3, particularly for the As vacancy, which has a higher probability of forming since the films were grown Ga-rich.

The Raman spectroscopy produced the results shown in Figure 6 for some of the samples. The literature shows two strong peaks for undoped GaAs: one at 292 cm^−1^, assigned as an LO phonon, and another at 268 cm^−1^, assigned as a TO phonon [37]. Additionally, weak responses can be distinguished in the region between 130 and 250 cm^−1^, corresponding to acoustic phonons [38], and two small peaks between 500 and 550 cm^−1^, called 2TO [39]. Figure 6a shows that in undoped GaAs, the TO peak is very small compared to the LO, but as the concentration grows, it gradually increases as a function of concentration. The reason for this is that the TO phonon is forbidden for a (100) geometry with normal backscattering of the Raman laser beam [40], as demonstrated in our measurements. Therefore, the appearance and increase of the TO peak indicate a loss of symmetry caused by impurities.

Furthermore, two contributions can be distinguished between the TO and LO phonons, which have been identified as surface-localized SO phonons [41] and the IFCM peak (Intermediate Frequency Coupled Mode) [42]. Nadaff [37] assigns a contribution given by As Vacancy at 250 cm^−1^. There is also a broad band region between 330 cm^−1^ and 500 cm^−1^, referred to in the literature as L+, which corresponds to transitions that involve the plasmon generated in the photon–electron interaction.

In Figure 7, the theoretical L+ plasmon scattering curve obtained from Espinoza’s thesis [43] is depicted. In this figure, we have overlaid the experimental values obtained by Espinoza for n-type GaAs. Similarly, we have superimposed the experimental values of the position of the maximum peak of the L+ band, based on the impurity concentration measurement made in 1995 and represented by red squares. For undoped GaAs, a native defect concentration of $1\times 10^{13}$ cm^−3^ was assumed, because that is the value that has been measured in undoped GaAs. This is indicated by a gray square in the figure. Clearly, our experimental results do not align with the theory in this manner. The discrepancy is due to the fact that carriers contributed by defects that have arisen due to aging were not taken into account. Adding a defect concentration of $7\times 10^{17}$ cm^−3^ to each of the impurity concentration values results in the outcome indicated by the black stars in the figure, which shows very good agreement.

The L+ band increases with impurities to the point that it can be higher than the LO peak at doping levels such as $1\times 10^{18}$ cm^−3^. This broad band indicates the presence of local vibration modes caused by defects in the lattice, and these local modes increase due to the aging process.

## 4. Conclusions

In this work, we found that O_2_ created oxides of As and Ga in GaAs epitaxial films preserved in a dry and dark environment and aged over a period of 26 years, causing vacancies of both elements, particularly of As. When comparing the shift of the deep defect band in the photoluminescence spectra over the years, we observed a correlation between the energy shifts of this defect band and the energy of the charge states of the As vacancy. Oxygen penetrates deep into the volume of the GaAs film, removing Ga and As atoms, indicating that this phenomenon is not just superficial. Through two independent experimental techniques, it was determined that the concentration of vacancies created over time can be as high as $1\times 10^{18}$ cm^−3^. We found that vacancies had a greater impact at low impurity concentrations, but as impurity levels increased, impurity-vacancy complexes are formed. These complexes caused greater deviations in the lattice parameter than external impurification itself. We identified two relaxation times for electrons in aged samples, which can be associated with electron de-excitation. One longer time is associated with the transition from the band to a deep vacancy trap level, while the shorter time is associated with the donor–acceptor transition between the impurity-vacancy complex.

## Figures and Tables

**Figure 1 materials-17-01399-f001:**
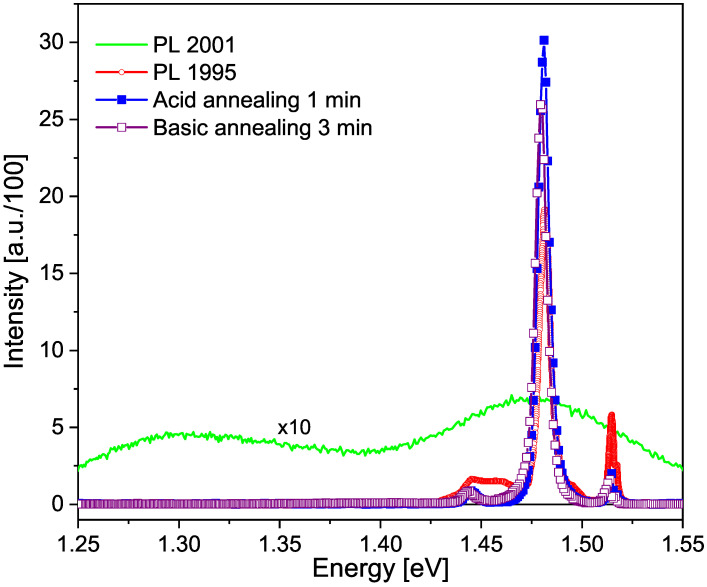
Photoluminescence spectra of a GaAs:Sn $5\times 10^{17}$ cm^−3^ sample, as described in the legend. Taken from [17] with the permission of G. Fonthal, one of the authors of this paper and G. Torres, who provided the GaAs samples.

**Figure 2 materials-17-01399-f002:**
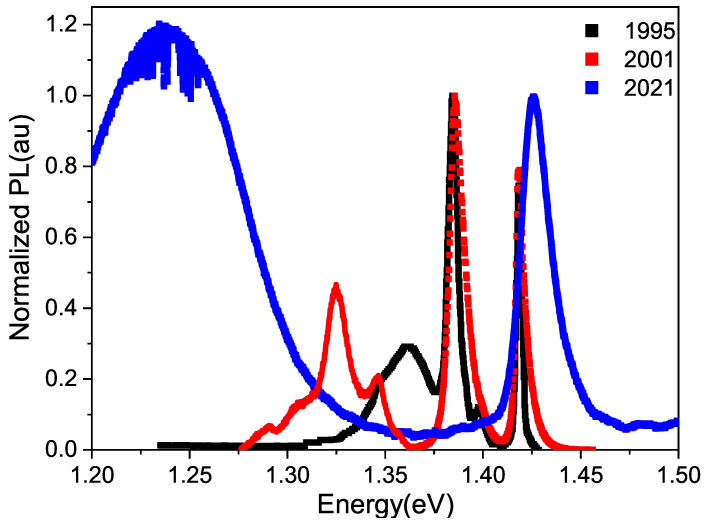
Normalized photoluminescence spectra obtained using a 488 nm laser at a temperature of 300 K, taken on different dates for an undoped GaAs sample. Spectrum in black taken from reference [15] and in red taken from [17].

**Figure 3 materials-17-01399-f003:**
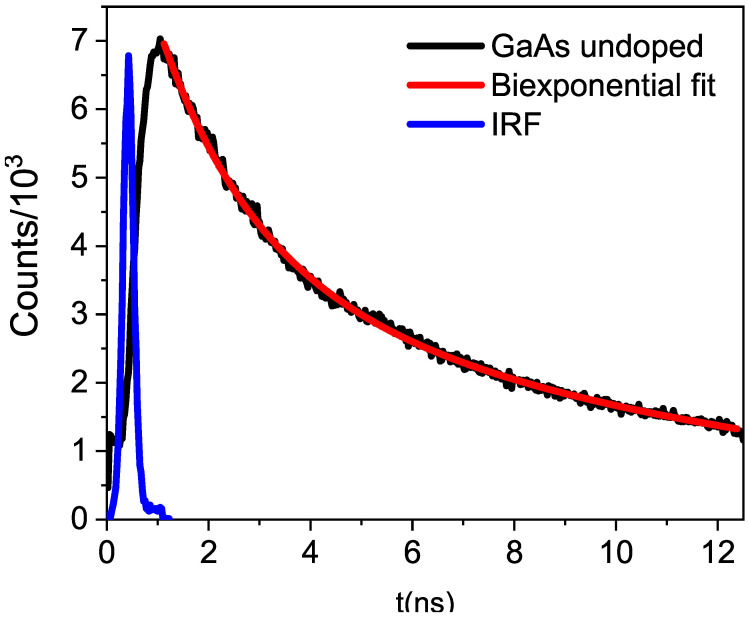
The TRPL response from an undoped GaAs sample (black line) is shown. The IRF (blue line) and the fit using a biexponential function (red line) are also displayed.

**Figure 4 materials-17-01399-f004:**
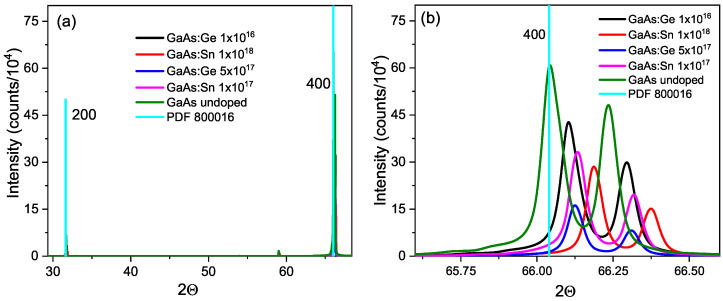
(**a**) X-ray diffractogram at 300 K for doped and undoped GaAs. (**b**) enlargement of peak on the (400) line.

**Figure 5 materials-17-01399-f005:**
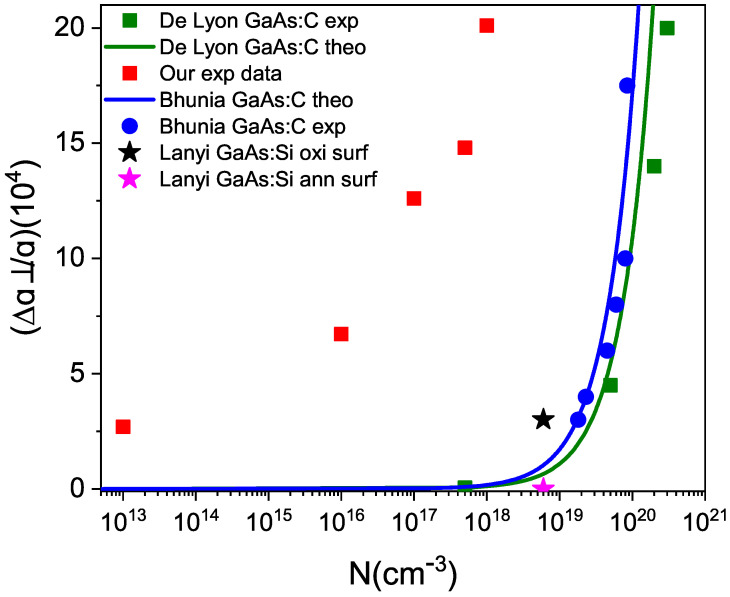
$△a_{⊥}/a$ plotted against the impurity concentration for the peak (400) of the GaAs diffractogram at 300 K, as reported by different authors. The references for the other authors can be found in the text.

**Figure 6 materials-17-01399-f006:**
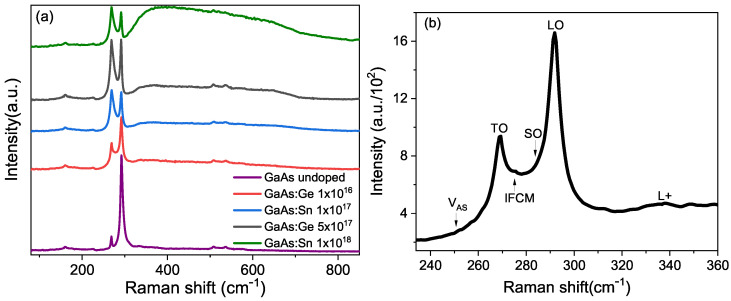
(**a**) Raman spectra at 300 K for some specified GaAs samples as shown in the box. (**b**) Raman spectrum for undoped GaAs showing the different contributions.

**Figure 7 materials-17-01399-f007:**
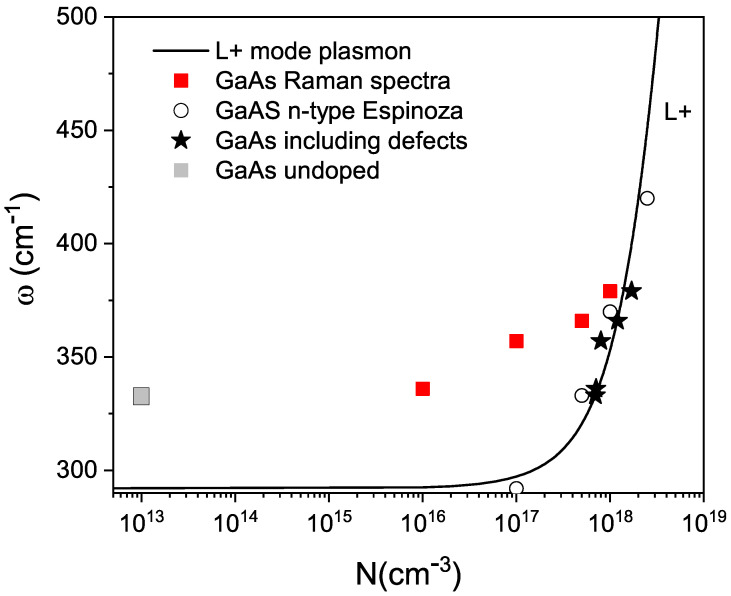
L+ plasmon scattering for the Raman spectra at 300 K for the GaAs samples shown in Figure 6. The explanation of the symbols is provided in the inset. Data is taken from Espinoza [43].

**Table 1 materials-17-01399-t001:** Relaxation times and participation percentage.

SAMPLE	t_1_	A1%	t_2_	A2%
GaAs undoped	0.80 (±0.10)	14	7.18 (±0.33)	86
GaAs:Ge $1\times 10^{16}$	1.74 (±0.16)	39	8.26 (±1.50)	61
GaAs:Sn $1\times 10^{17}$	1.71 (±0.13)	41	8.60 (±1.50)	59
GaAs:Ge $5\times 10^{17}$	1.67 (±0.10)	44	9.56 (±1.89)	56
GaAs:Sn $1\times 10^{18}$	1.47 (±0.04)	45	9.76 (±2.11)	55
GaAs:Ge $1\times 10^{19}$	1.86 (±0.14)	61	9.10 (±2.16)	39

**Table 2 materials-17-01399-t002:** Lattice parameters in the (100) direction, perpendicular, and grain size.

SAMPLE	*a* (Å)	$△a_{⊥}/a\left(10^{-3}\right)$	D (μm)
GaAs undoped	5.6528	0.27	0.130
GaAs:Ge $1\times 10^{16}$	5.6498	0.67	0.141
GaAs:Sn $1\times 10^{17}$	5.6478	1.26	0.134
GaAs:Ge $5\times 10^{17}$	5.6472	1.48	0.188
GaAs:Sn $1\times 10^{18}$	5.6431	2.01	0.182

**Table 3 materials-17-01399-t003:** Vacancies energies of As and Ga according to their state of charge [34].

Vacancies with Its State of Charge	Trap Energy in eV
$V_{Ga}^{+1}$	0.100
$V_{Ga}^{+2}$	0.068
$V_{Ga}^{+3}$	0.051
$V_{As}^{-1}$	0.145
$V_{AS}^{-2}$	0.088
$V_{AS}^{-3}$	0.042

## Data Availability

No new data were created or analyzed in this study. Data sharing is not applicable to this article.

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
