# Peer review of "Formation of Point Defects Due to Aging under Natural Conditions of Doped GaAs"

_materials, 2024, doi:10.3390/ma17061399_

Round 1
Reviewer 1 Report
The authors investigated the aging dynamics of GaAs doped with Ge and Sn. The work demonstrated photoluminescence, Raman, and X-ray measurements in three periods: 1995, 2001, and 2021. The work found that oxygen formed oxides that gave off Ga and As atoms, leaving vacancies mainly of As. There are several issues that should be addressed before the consideration of publication.
1. The work tested for 26 years to clarify the aging dynamics of under natural condition of doped GaAs, that is O2 created oxides of As and Ga, particularly of As. Is the result different from the aging experiment for short periods?
2. In this work, the samples were stored in dark and dry environment for long periods. However, experimental samples were normally stored in vacuum condition to prevent surface oxidation and interaction. Please state the significance of the results from the sample exposed to natural condition.
3. In Fig. 2, the PL from 1995 to 2001, the peak at 1.36 eV disappeared, from 1995 to 2021, the broad peak at 1.24 emerged. Please state the reason for the changes.
4. Please state how the experimental results in this work can be applicable to the GaAs-based microelectronic devices including solar cell and transistor as stated in the introduction.
Reviewer 2 Report
The manuscript investigates the properties of GaAs substrates over a period of 26 years, using various techniques such as PL, Raman, etc. The study reports on the formation of vacancies in the epitaxial layer and their impact on impurity levels. While the topic is intriguing, this manuscript requires significant improvement, particularly in experimental design and scientific writing. The authors should aim to delve deeper into the findings and extract more insights to showcase the novelties of their work. Here are some suggestions that the authors can work on to further enhance this article:
1. In line 3, I believe there is a typo error, and it should be 'Photoluminescence' instead of 'Foltoluminescence’.
2. In Line 27, it should be 'GaAs-based' instead of 'Gas-based’
3. From line 43 to line 46, it is not necessary to explain each section in detail. A brief mention of the paper's motivation, study, or focus should be sufficient in this part.
4. In line 50, could you kindly clarify the meaning of the notation "7N, 5N, 5N" in the brackets?
5. In line 51, please provide a reference to support the statement "As can be seen, the resulting films will contain native VAS...".
6. In line 52, please add annealing details, including the annealing temperature, duration, and atmosphere used during the process.
7. For Figure 1, it appears to be a reprint image. As such, please provide a revised description that includes the appropriate copyright information for the reprint. Additionally, there is a typo error in the legend, which reads "anneling"; kindly correct it to "annealing."
Furthermore, I noticed that the authors have also reprinted other work in the manuscript. Please ensure that you obtain the necessary permissions and copyrights for these reproductions. Make the required modifications and provide proper acknowledgments for any reprinted material.
8. In line 100, please specify the ambient conditions under which the experiments were conducted.
9. For Figure 2, if you are using spectra from other researchers' work, it is essential to cite their work appropriately. Please ensure that you provide proper citations both within the figure itself and in the figure description.
10. In line 105, it is well known that the direct bandgap energy of GaAs is 1.424 eV (corresponding to ~870 nm). I believe the value of 1.375 eV mentioned in the manuscript for the band-to-band transitions is a typo error. Please correct it.
11. In line 107, regarding the discussion of the peak shifting during aging, I observed that the peak shifts to a higher energy from the figure 2, contrary to what was mentioned in the manuscript. Please clarify the reason for this discrepancy and provide an explanation for the observed peak shift to a higher energy during aging. Additionally, in Figure 2, there is a peak observed at 1.25 eV for the blue curve. Please give an explanation for the origin of this peak in the figure.
12. Line 230, it’s figure 7 not 6.
Please review and revise the manuscript to address typographical errors, grammatical mistakes, and ensure clear scientific writing. Additionally, provide accurate citations and permissions for external material used
Round 2
Reviewer 1 Report
The authors addressed the inquires from the reviewer.
Some typos need to be checked.
Reviewer 2 Report
I believe the author has addressed all the issues and questions I raised in my previous review. The paper has been significantly improved. I recommend accepting this manuscript for publication in this journal.